# Stacking Multiple Genes Improves Resistance to *Chilo suppressalis*, *Magnaporthe oryzae*, and *Nilaparvata lugens* in Transgenic Rice

**DOI:** 10.3390/genes14051070

**Published:** 2023-05-12

**Authors:** Bai Li, Zhongkai Chen, Huizhen Chen, Chunlei Wang, Liyan Song, Yue Sun, Yicong Cai, Dahu Zhou, Linjuan Ouyang, Changlan Zhu, Haohua He, Xiaosong Peng

**Affiliations:** 1Key Laboratory of Crop Physiology, Ecology and Genetic Breeding, Ministry of Education, Research Center of Super Rice Engineering and Technology, Jiangxi Agriculture University, Nanchang 330045, China; lb15288316292@163.com (B.L.);; 2Pingxiang Center for Agricultural Sciences and Technology Research, Nanchang 330200, China

**Keywords:** transgenic rice, *Chilo suppressalis*, *Magnaporthe oryzae*, *Nilaparvata lugens*, rice blast

## Abstract

The ability of various pests and diseases to adapt to a single plant resistance gene over time leads to loss of resistance in transgenic rice. Therefore, introduction of different pest and disease resistance genes is critical for successful cultivation of transgenic rice strains with broad-spectrum resistance to multiple pathogens. Here, we produced resistance rice lines with multiple, stacked resistance genes by stacking breeding and comprehensively evaluated their resistance to *Chilo suppressalis* (striped rice stemborer), *Magnaporthe oryzae* (rice blast), and *Nilaparvata lugens* (brown planthopper) in a pesticide-free environment. *CRY1C* and *CRY2A* are exogenous genes from *Bacillus thuringiensis*. *Pib*, *Pikm,* and *Bph29* are natural genes in rice. CH121TJH was introduced into *CRY 1C*, *Pib*, *Pikm,* and *Bph29*. CH891TJH and R205XTJH were introduced into *CRY 2A*, *Pib*, *Pikm,* and *Bph29*. Compared with those observed in their recurrent parents, CH121TJH significantly increased the mortality of borers. The other two lines CH891TJH and R205XTJH are the same result. Three lines introduction of *Pib* and *Pikm* significantly reduced the area of rice blast lesions, and introduction of *Bph29* significantly reduced seedling mortality from *N. lugens*. Introduction of the exogenous genes had relatively few effects on agronomic and yield traits of the original parents. These findings suggest that stacking of rice resistance genes through molecular marker-assisted backcross breeding can confer broad spectrum and multiple resistance in differently genetic backgrounds.

## 1. Introduction

Rice is an important staple crop whose production is crucial to global food security [1]. In current agricultural practice, diseases and insect pests are significant challenges to safe and sustainable rice production, and reducing their impact is critical for ensuring continued food security [2]. *Chilo suppressalis* is an economically important pest, which can cause serious damage to rice. It is estimated that the loss caused by *C. suppressalis* accounts for 10–30% of total rice production [3]. In many rice-growing countries in Asia and Africa, rice blast has re-emerged as the main factor affecting the stable production of rice and food security. For example, in the highlands of eastern India alone, 50% of rice production is lost on average every year [4]. From 2005 to 2006, serious pests occurred in southern Vietnam. More than 485,000 hectares of rice fields were affected by viral diseases that seemed to be transmitted by *N. lugens* and other insect vectors, resulting in a loss of 120 million US dollars [5]. At present, chemical prevention is still the main method of ensuring stable yields; however, long-term pesticide use, especially the excessive use of chemical pesticides, has enabled many pests to develop resistance to commonly used chemical insecticides and fungicides [6]. Cloning disease and pest resistance genes and incorporating these genes into new rice varieties through molecular breeding is therefore crucial for protecting the environment and promoting sustainable agricultural development [7].

Most cultivated rice varieties were bred for strong single resistance traits. However, in actual production, a variety of diseases and pests typically occur simultaneously or successively [8]. To prevent loss of resistance after large-scale planting of resistant varieties, it is necessary to further expand the sources of resistance genes, express resistance genes with different resistance spectra, and develop multi-gene-resistance varieties with strong field adaptability. Many breeders have therefore begun to combine multiple resistance genes in the same elite variety in hopes of extending its useful lifespan [9]. Aggregating different resistance genes into one variety can not only further improve its resistance level but also enhance its broad-spectrum and durable resistance. Therefore, aggregation of two or more resistance genes into the same variety is an important breeding goal [10].

Gene stacking breeding involves the collection and stacking of target genes, followed by the fixation of polymeric genes, that is, their homozygous stacking [11]. There are two basic approaches to plant molecular breeding by gene stacking [12]. The first is molecular breeding by gene stacking and genetic transformation. This method combines conventional breeding techniques with plant transformation methods such as plant DNA virus–mediated transformation, *Agrobacterium*-mediated transformation, microinjection, electric shock, and pollen tube pathway–mediated transformation. Such strategies are used to transfer two or more artificially modified and isolated genes into the recipient plant. Expression of the imported genes causes heritable modifications of plant characteristics, enabling the creation of new plant varieties with specific target traits. The second approach involves gene aggregation by molecular marker–assisted selection. In this method, favorable genes are aggregated into the same genome through techniques such as pair hybridization, synthetic hybridization, additive hybridization, multi-paternity mixed pollination hybridization, and backcrossing. Using molecular markers to select a single plant with multiple target genes from offspring, and then select highly resistant strains to realize the stacking of favorable genes.

A number of studies in rice and maize demonstrate the possibilities of gene stacking breeding. Sequence-tagged site (STS) marker-assisted selection was used to pyramid two bacterial blight resistance genes, *Xa21* and *Xa13*, and one aroma gene, *(fgr)*, into multiple transgenic rice lines. In the BC_2_F_3_ and BC_3_F_3_ generations, the transgenic plants exhibited a wider resistance spectrum and stronger resistance than plants with a single resistance gene [13]. Similarly, the rice blast resistance genes *Pi-ta*, *Pi-b*, and *Pi-d(t)* from Pi-4, BL-1, and Digu were polymerized into G46B, significantly improving its blast resistance [14]. Molecular marker-assisted selection and backcross breeding techniques can therefore effectively improve rice resistance.

The use of diverse resistance genes is important for coping with variations among biotypes or species of pests and pathogens and for enhancing the breadth and durability of plant resistance [15]. In this study, resistance genes carried in rice lines MH63, CK30, and PSL were transferred into three rice restorer lines, CH121, CH891, and R205X, by molecular marker-assisted selection and backcrossing techniques. A group of polygenic improved lines carrying resistance genes to *C. suppressalis* (striped rice stemborer), *Magnaporthe oryzae* (rice blast), and *N. lugens* (brown planthopper) were identified, and their resistance levels and agronomic characteristics were comprehensively evaluated.

## 2. Materials and Methods

### 2.1. Plant Materials

The donor parents were MH63(*CRY1C*), MH63(*CRY2A*), CK30, and PSL. MH63(*CRY1C*) and MH63(*CRY2A*), which carry the *C. suppressalis* resistance genes *CRY1C* and *CRY2A*, were provided by Huazhong Agricultural University. Appendix A shows the information of transgenes-*CRY 1C* and *CRY 2A*. CK30 carries the *Pibgom* and *Pikm* rice blast resistance genes and was provided by Sichuan Agricultural University. PSL carries the *N. lugens* resistance gene *Bph29* and was provided by Chengdu Institute of Biology, Chinese Academy of Sciences. The acceptor parents were CH121, CH891, and R205X, high-quality restorer lines bred by Jiangxi Agricultural University. They have good adaptability in the middle and lower reaches of the Yangtze River, the main rice producing area in China. Descriptions of plant materials description are shown in Table 1.

Hybridization and breeding were performed at the transgenic experimental base of Jiangxi Agricultural University, Nanchang City, Jiangxi Province beginning in 2010. With two seasons per year, we obtained stable genetic strains by 2020. At the beginning of the experiment, the upper 15 cm of soil at the test site had the following properties: pH 5.01, 1.26 g kg^−1^ total N, 105.6 mg kg^−1^ available P, 125.2 mg kg^−1^ K, and 20.56 g kg^−1^ organic matter. The study was conducted at the Transgenic Test Base of Jiangxi Agricultural University (28°48′10″ N, 115°49′55″ E) in the economic and technological development zone, Nanchang City, Jiangxi Province during the 2020 rice growing season.

For *CRY1C*, CH121 was the recipient parent, and MH63(*CRY1C*) carrying the *CRY1C* borer resistance gene was the donor parent. For *CRY2A*, CH891 and R205X were the recipient parents, and MH63(*CRY2A*) carrying the *CRY2A* borer resistance gene was the donor parent. Through backcrossing, heterozygous single plants carrying the target genes were screened out from the BC_1_F_1_ generation. Among these heterozygous single plants, individuals with a leaf morphology similar to that of their respective recurrent parent were further screened out until the BC_3_F_1_ generation. The BC_3_F_2_ population obtained after self-crossing was screened using molecular markers, and excellent single strains carrying homozygous target genes were selected. The BC_3_F_4_ population was obtained by self-crossing, and a stable line with a leaf morphology similar to the recurrent parent and the insect resistance gene *CRY1C* was selected and named CH121T. The CH891T and R205XT lines containing the *CRY2A* gene were produced using the same process. CH121T, CH891T, and R205XT were then used as the receptor parents and CK30 as the donor parent to obtain lines with the rice blast resistance genes *Pib* and *Pikm* and agronomic traits similar to those of their parents; these lines were named CH121TJ, CH89TJ1, and R205XTJ. Similarly, *Bph29* was added to CH121TJ, CH89TJ, and R205XTJ by backcrossing with PSL. The polymeric resistance gene varieties CH121TJH (containing *CRY1C*, *Pib*, *Pikm*, and *Bph29*) and CH89TJH and R205XTJH (containing *CRY2A*, *Pib*, *Pikm*, and *Bph29*) were obtained by self-crossing and provided new genetic materials for subsequent resistance experiments.

### 2.2. Molecular Marker-Assisted Detection

Genomic DNA was extracted from the parent varieties and the new varieties with polymerized resistance genes by the CTAB method [16]. The parent varieties were MH63(*Cry1C*), MH63(*Cry2A*), CK30, PSL, CH121, CH891, and R205X, and the varieties with polymerized resistance genes were CH121TJH, CH89TJH, and R205XTJH. DNA concentration and purity were assessed using a NanoDrop 2000 spectrophotometer (Thermo Scientific) based on the absorbance ratio of 260/280 nm [17]. The diluted DNA was amplified by PCR using the primer sequences shown in Appendix A. The PCR products were separated and visualized by agarose gel electrophoresis. The band sizes of *CRY1C* and *CRY2A* were 799 bp and 600 bp. The *Pikm* blast resistance gene is composed of two closely related, adjacent genes, *Pikm1* and *Pikm2*. They were amplified using the specific primers DKM1 and DKM2 and had band sizes of 191 bp and 223 bp. Previous work has shown that blast resistance requires the presence of both bands in a single plant [18]. Primers for *Pib* included *Pib*, which detects the presence of the resistant allele (band size 360 bp) and *Lys145*, which detects the presence of the susceptible allele (800 bp). The appearance of both bands indicated that the tested material was heterozygous [19]. The band produced by amplification of the *N. lugens* resistance gene *Bph29* was 570 bp.

### 2.3. Genetic Background Detection Based on SSR Markers

Whole genomes of the recurrent parents CH121, CH891, and R205X and the corresponding new strains CH121TJH, CH891TJH, and R205XTJH were analyzed using 560 pairs of SSR primers evenly distributed on the 12 rice chromosomes. Polymorphic primer markers were identified, and the genetic background recovery rate was analyzed. The SSR primer sequence reference (http://www.gramene.org/microsat/ accessed on 1 October 2021.) was synthesized by Beijing Tsingske Biotechnology Company. Diluted genomic DNA was used for PCR amplification, and the PCR products were separated by SDS–PAGE on an 8% polyacrylamide gel with silver staining. The physical locations of molecular markers are shown in Appendix A. A single plant with a homozygous band pattern was marked as 1, a single plant with a heterozygous band pattern was marked as 2, and a single plant with no bands was marked as 0.

The numb of that polymorphic molecular markers screen out between the donor parent and each corresponding parent is L, and further screening is carried out between the correspond parent and the polymeric resistance gene strain to obtain the number of the molecular markers of the same band type between the corresponding polymeric resistance gene strain and the recurrent new strain is X. According to the actual recovery rate formula [20]:R[G(g)] = [L + X]/2L,

R[G(g)] is actual recovery rate after backcrossing g generations. The actual recovery rate of that rice plant line with the stacking resistance gene can be calculated (Appendix A) through one-time hybridization, three-time backcross, and three-time selfing, according to a theoretical recovery rate formula:G(g) = 1 − (1/2)^g+1^

G(g) is theoretical recovery rate after backcrossing g generations. The theoretical recovery rate of each transgenic line after the introduction of the *CRY1C* or *CRY 2A* gene was calculated to be 93.75%; Through one-time hybridization, three times of backcross and three times of selfing, the theoretical recovery rate of each transgenic plant line after the rice blast resistance gene is introduced is calculated to be 88.25%; and finally, after the *Bph 29* gene is introduced through one-time hybridization, three times of backcross and three times of selfing, the theoretical recovery rate of each transgenic plant line after the anti-rice blast gene is introduced is calculated to be 83.34%, and the recovery rate is the final theoretical recovery rate of the polymerized gene rice.

### 2.4. Insect Resistance in the Laboratory

Insect feeding assays were performed using 10 replicates of each transgenic line and non-transgenic control. In August 2021, female *C. suppressalis* were captured using light in the experimental field of Jiangxi Agricultural University, then maintained at 28 °C under humid conditions for spawning. When black spots gradually appeared in the eggs, they were transferred to fresh stalks of rice without *C. suppressalis* resistance, and the larvae were allowed to feed for about 10 days [21]. Second instar larvae were fed fresh transgenic rice leaves and stems that had been collected from rice plants at the tillering and heading stages; non-transgenic rice tissues at the same growth stages were used as negative controls. Second instar larvae were placed in individual Petri dishes that contained a piece of leaf (4 g) or stem (5 g), and ddH_2_O was added to the filter paper to maintain humid conditions. The initial number of larvae per Petri dish was 30. Petri dishes were sealed with parafilm membranes to prevent larvae from escaping. All Petri dishes were stored in a hermetic box in the dark at 27 ± 1 °C and 75 ± 5% relative humidity [22]. Larval mortality was determined after 48 h, and the mortality rate was calculated. The calculations [23] and statistical methods are shown in Appendix A.

Statistical equation of larval mortality:larval mortality = (total number of larval − alive number of larval)/total number of larval × 100%

Statistical equation of corrected larval mortality:corrected larval mortality = (larval mortality of transgenic rice − larval mortality of negative controls rice)/(1 − larval mortality of negative controls rice) × 100%

### 2.5. Assessment of Rice Blast Resistance at the Seedling Stage

The control materials were CK30 (with high rice blast resistance), LTH (with high rice blast sensitivity), CH121, CH891, and R205X. The experimental materials were the new lines CH121TJH, CH891TJH, and R205XTJH. Two strains of *M. oryzae* were isolated and purified from a field in Jiangxi Province by researchers from Jiangxi Academy of Agricultural Sciences and named F1-6 and 253. PDA medium was used as the activation medium and rice bran medium as the sporulation medium (formulas shown in Appendix A). Spores of *M. oryzae* were activated on PDA medium for 7 days and then transferred to rice bran medium. The two strains were cultured separately. After inoculation, both media were cultured in an incubator at 28 °C. After mycelium overgrew on the rice bran medium plate (about 7 days), it was placed on a tissue culture rack at 25–28 °C for 24 h of light culture. After full sporulation (about 5 days), conidia of *M. oryzae* were eluted with 0.02% TWEEN-20; this procedure was repeated 2–3 times, and the eluant was filtered through sterilized gauze. The filtered spore suspensions were counted at 100× under a microscope and inoculated when there was an average of 30–50 spores per field of view [24]. The concentration of spores was about 30 × 10^4^ conidia/mL.

Seeds of the experimental and control groups were immersed in clear water at 25 °C for 48 h, germinated at 35 °C, and then sown in seedling trays. When rice seedlings had grown to the 3–4 leaf stage, spray inoculation was performed in an inoculation box using an airbrush with pressurized air. Spore suspension (40 mL) was sprayed onto each seedling tray, and there were three replicate trays per genotype and treatment. Control trays were sprayed with water only. After inoculation, trays were maintained in a dark, humid incubator at 28 °C. The incidence of rice blast was investigated 10 days after inoculation; incidence, lesion size, and lesion number were recorded on the basis of visual observation. The disease condition was recorded, and the resistance level was determined according to the six-point scale of Mackill et al. [25]. The identification criteria are shown in Appendix A.

### 2.6. Identification of Resistance to N. lugens at the Seedling Stage

The Standard Seedbox Screening Technique (SSST) [26] was used to determine the resistance of rice seedlings to *N. lugens*. The mixed biotype insect source was collected from the experimental field of Jiangxi Agricultural University and was fed with conventional rice that lacked the *N. lugens* resistance gene. Seeds of new materials (CH121TJH, CH891TJH, and R205XTJH), recurrent parents (CH121, CH891, and R205X), and control materials (susceptible TN1 and resistant PSL) were immersed in clear water at 25 °C for 48 h, germinated at 35 °C, and then sown in seedling trays. Twenty-eight seeds were sown in each tray, and there were three replicate trays per genotype. *N. lugens* was added when the plants reached the three-leaf stage. Six to seven nymphs of 2–3 instars of *N. lugens* were added to each plate and allowed to feed and breed. When almost all TN1 plants had withered, resistance was scored on the basis of damage symptoms using the criteria shown in Appendix A.

### 2.7. Measurement of Agronomic Traits

Transgenic rice lines were planted in paddy fields at the Transgenic Experimental Plots of Jiangxi Agricultural University (Nanchang, Jiangxi, China) to evaluate their agronomic performance. Non-transgenic lines CH121, CH891 and R205X, were planted in paddy fields adjacent to the transgenic lines as controls. We used pesticides and fungicides in the cultivation process in order to create an environment to control pests and diseases. The purpose of this is to examine whether there is a huge yield difference between the transgenic rice lines and their corresponding parents under normal conditions without pests and diseases. Eight yield traits were measured: plant height, total number of grains, effective panicle number, panicle length, seed-setting rate, yield per plant, filled grain number per panicle, and 1000-grain weight. Filled and unfilled grains of the main panicle were separated manually for measurement of seed-setting rate (filled grains/(filled grains + unfilled grains) × 100). Yield per plant was calculated as panicles per plant × grains per panicle × 1000-grain weight × seed-setting rate × 10^−6^. Details are shown in Appendix A. Agronomic traits of transgenic plants and the recurrent parents were compared using one-way analysis of variance (ANOVA), and values are presented as means ± SD. Figures were constructed using Origin 2017 (OriginLab Corp., Northampton, MA, USA).

## 3. Results

### 3.1. Resistance Gene Stacking Analysis

The PCR amplification products of the donor parents, recurrent parents, and lines with polymerized resistance genes were analyzed by agarose gel electrophoresis. The target fragments of the borer resistance genes *CRY1C* and *CRY2A* were successfully amplified from the three polymerized resistance gene lines and the respective positive control lines MH63(*Cry1C*) and MH63(*Cry2A*) (Figure 1A). The target fragment of *Pib* was successfully amplified in the three polymerized resistance gene lines and the positive control CK30. By contrast, *Lys145* was not amplified in the three new lines but was amplified in the control CH121 (Figure 1B). The target fragment of the brown planthopper resistance gene *Bph29* was successfully amplified in the three polymerized resistance gene lines and the positive control PSL but not in the negative control CH121 (Figure 1C). The target fragments of the *PiKm* blast resistance genes were amplified in the three transgenic rice cultivars and the positive control CK30 (Figure 1D). Molecular marker-assisted detection indicated that *CRY1C*, *Pib*, *Pikm*, *Bph29*, and *Pib* were homozygous in CH121TJH plants. *CRY2A*, *Pib*, *Pikm*, and *Bph29* resistance genes were present in CH891TJH and R205XTJH plants, and *Pib* was homozygous. Thus, after multiple generations of backcrossing and self-cross breeding (Figure 2), resistance genes for *C. suppressalis*, *M. oryzae*, and *N. lugens* were successfully introduced into CH121TJH, CH891TJH, and R205XTJH.

### 3.2. Analysis of Genetic Background Recovery Rate

PSL parent lines and CH121, CH891, and R205X were tested with 560 pairs of microsatellite markers, and the numbers of polymorphic markers were 53, 49, and 55. The rates of polymorphism were 9.46%, 8.75%, and 9.82% (Figure 3). The actual recovery rate of the parental genetic background was 85.85% for CH121 and CH121TJH, 90.82% for CH891 and CH891TJH, and 81.82% for R205X and R205XTJH. The actual recovery rates of CH121TJH and CH891TJH were higher than the theoretical rate of 83.34%, whereas that of R205XTJH was lower (Appendix A).

### 3.3. Assessment of Resistance to C. suppressalis

In the test for *C. suppressalis* resistance, the number of dead borers on CH121, CH891, and R205X ranged from 1.33 to 2.00, and the borer mortality rate ranged from 4.89% to 6.74% (Figure 4A, Appendix A). By contrast, the number of dead borers on CH121TJH, CH89TJH, and R205XTJH ranged from 24.67 to 26.67, and the mortality rate ranged from 82.23% to 88.90%. The corrected mortality rates were 88.16%, 82.57%, and 82.23% for CH121TJH, CH89TJH, and R205XTJH, respectively (Appendix A). On the basis of their *C. suppressalis* resistance, CH121, CH891, and R205X were classified as Sensitive (S), CH121TJH as Highly Resistant (HR), and CH89TJH and R205XTJH as Resistant (R) (Figure 4B, Appendix A). Thus, introduction of the borer resistance gene *CRY1C* or *CRY2A* increased the resistance of rice plants to borer larvae compared with that of their original parents, and resistance reached the HR level in CH121TJH.

### 3.4. Assessment of Rice Blast Resistance at the Seedling Stage

After they were sprayed with the rice blast strains F1-6 and 253, leaves of the susceptible control LTH exhibited typical fusiform rice blast lesions with a diameter of >3 mm. The lesions fused into patches, causing the tops of the leaves to die, thereby classifying LTH as HS. By contrast, leaves of CK30 showed no lesions, classifying CK30 as HR. Leaves of CH121, CH891, and R205X exhibited typical fusiform rice blast lesions with diameters of 1–3 mm. The lesions were slightly fused, classifying these lines as S. CH121TJH showed no leaf lesions and was therefore classified as HR. Both CH891TJH and R205XTJH exhibited sporadic brown spots <0.5 mm in diameter and were therefore R (Figure 5A,B). All data on infected leaves are presented in Appendix A.

### 3.5. Assessment of N. lugens Resistance at the Seedling Stage

In the test of resistance to *N. lugens*, the number of dead TN1 seedlings was 26.67 ± 1.25, and the mortality rate of TN1 was 94.05 ± 3.37%, indicating that this line was HS. No PSL seedlings died after exposure to brown planthopper, indicating that PSL was Immune (I). The number of dead seedlings of CH121 was 24.67 ± 2.49, and the mortality rate was 88.09 ± 8.91%, classifying CH121 as HS. There were 18.33 ± 1.25 (65.48 ± 4.46%) dead seedlings in CH891 and 14.33 ± 1.25 (51.19 ± 4.45%) in R205X, indicating that both of these varieties were S. The number of dead seedlings of CH121TJH was 2.33 ± 2.05, and the mortality rate was 8.33% ± 2.34%, classifying CH121TJH as HR. There were 5.33 ± 1.25 (19.05 ± 4.45%) dead seedlings in CH891TJH and 7.33 ± 1.25 (26.19% ± 4.45%) in R205X, indicating that both of these varieties were R (Figure 6, Appendix A). Thus, introduction of the *Bph29* resistance gene increased the *N. lugens* resistance of three transgenic rice lines compared with that of their original parents, and CH121TJH was highly resistant.

### 3.6. Analysis of Agronomic Traits

We next measured eight agronomic traits of the three parents (CH121, CH891, and R205X) and the three polymerized resistance gene lines (CH121TJH, CH891TJH, and R205XTJH) under field conditions. There were no significant differences (*p* > 0.05, LSD test) in plant height, effective panicle number, panicle length, and seed setting rate between the three transgenic rice lines and their parents (Figure 7A,C–E). CH121TJH and R205XTJH had slightly lower total grain number than their corresponding parents (*p* < 0.01), but there was no significant difference in grain number between CH891TJH and CH891 (*p* > 0.05) (Figure 7B). R205XTJH had lower yield per plant and 1000-grain weight than its parent R205X (0.01 < *p* < 0.05), but CH121TJH and CH891TJH did not differ significantly from their parents in these parameters (*p* > 0.05) (Figure 7F,H). Number of filled grains per panicle was also lower in R205XTJH than in its parent (*p* < 0.01), but CH121TJH and CH891TJH did not differ from their parents (*p* > 0.05) (Figure 7G). Thus, with the exception of R205XTJH, which had somewhat lower total grain number, filled grain number per panicle, and 1000-grain weight, most important agronomic traits were not significantly affected by the introduction of foreign genes in the other two newly created lines.

## 4. Discussion

Diseases and pests often occur simultaneously or in succession during rice production, limiting rice yield and threatening food security [27]. Here, we used marker-assisted backcross breeding (MABB) to aggregate the stemborer resistance gene *CRY1C*/*CRY2A*, the rice blast resistance genes *Pikm* and *Pib*, and the brown planthopper resistance gene *Bph29* into three high-quality indica rice restorer lines, obtaining the new lines CH121TJH, CH891TJH and R205XTJH. In previous research, researchers have typically aimed to optimize a plant’s resistance to a specific resistance gene by introducing multiple resistance genes. Selected three broad spectrum potato R genes (*R*pi), *Rpi-sto*1 (*Solanum stoloniferum*), *Rpi-vnt1.1* (*S. venturii*), and *Rpi-blb3* (*S. bulbocastanum*), combined into a single binary vector pBINPLUS and transformed into the susceptible cultivar Desiree. Through genetic transformation, potato had three R genes that are naturally resistant to single disease. [28]. Two antifungal genes were stacked into transgenic pea (*Pisum sativum L.*) to enhance resistance against fungal diseases, which proved the stable inheritance of the antifungal genes in the transgenic plants [29]. Compared with the technology aiming at a single disease gene, introducing more than one gene into crop simultaneously or sequentially (called transgenic stacking) is a more effective strategy to endow transgenic plants with higher and more lasting insects and disease resistance [30]. Two *Bacillus thuringiensis* (Bt) insecticidal genes, *Cry 1Ac* and *Cry 1Ig*, and a modified glyphosate-tolerant 5-enolpyruvylshikimate-3-phosphate synthase (*EPSPS*) gene (*G 10*) were combined into Elite rice (*Oryza sativa* spp. *japonica*) cultivar Xiushui 134, was found to be highly resistant to striped stem borer and rice leaf roller, and tolerant to glyphosate [31]. Three insecticidal genes (the Bt gene *Cry 1Ac* and *Cry 2A*, and the snowdrop lectin gene *gna*) were introduced into important commercial indica rice varieties M7 and Basmati370 at the same time. The bioassay using triple transgenic plants showed that rice leaf roller and yellow rice borer were completely eradicated, and the survival rate of brown planthopper was reduced by 25%. The greatest decrease in insect survival rate and plant damage occurred in plants expressing all three transgenes lines [32]. It is effective and feasible to stack R. gene into plants to enhance disease resistance and insect resistance. However, too much introduction of transgenes will also lead to unintended effects. Such as statistically significant differences in the phenotype, response, or composition of the GM plant compared with the parent from which it is derived [33]. Therefore, if more cis-genes can be used instead of transgenes, the risk of these unintended effects and unknown mutations will be reduced. Some scholars believe that cis-genes does not add an extra trait [34].

Molecular marker-assisted detection is a commonly used method in MABB. By comparing the size of a target gene amplified by specific primers between the parent and the progeny line, it is possible to determine whether the target gene has been successfully introduced [35,36]. MABB has previously been used to introduce *crtRB1* and *o2* into maize (*Zea mays* L.) to increase β-carotene, lysine, and tryptophan levels. Likewise, genes for tolerance/resistance to submergence (*Sub1*), salinity (*Saltol*), rice blast (*Pi2*, *Pi9*), and gall midge (*Gm1*, *Gm4*) were identified and introduced into the rice cultivar Tapaswini, which already contained four bacterial blight resistance genes [37]. In the present study, insect and blast resistance genes were detected using molecular markers, thereby enabling their aggregation into new rice varieties via backcross breeding, consistent with the approaches used in previous studies.

In crop backcross breeding, genetic background recovery rate is an important standard for measuring the similarity of background traits between parents and offspring [38], and genic microsatellite markers are commonly used for this purpose [39]. The *Sub1* site of FR13A, a submergence-tolerant variety, was introduced by MABB into CO 43, a rice variety prevalent in southern India. Genotyping and phenotyping of the BC_3_F_3_ generation showed that high-quality, near-isogenic lines of CO 43 contained the *Sub1* locus and had 94.37–95.78% of the recurrent parent CO 43 genome [40]. In another study, the rice variety MR219 was used as the recurrent parent, and Pongsu Seribu 1 carrying the rice blast resistance genes *Piz*, *Pi2*, and *Pi9* was used as the resistance gene donor. Using seventy microsatellite markers, the recovery rate of the recurrent parent genome in the improved lines was determined to be 95.98–97.70% [41]. After continuous backcrossing and self-crossing for multiple generations, the genetic background recovery rate of offspring of different recurrent parents will be different, even with the same donor parent. As expected, the actual recovery rates differed among the three rice lines created here; some were higher than the theoretical recovery rate, and some were slightly lower.

The presence of R gene(s) is the key to the resistance of resistant plants. After introduction of the stemborer resistance gene *CRY1C*, the rice blast resistance genes *Pib* and *Pikm*, and the brown planthopper resistance gene *Bph29*, CH121TJH showed significantly improved resistance to borers, rice blast, and brown planthopper compared with its parent CH121. The resistance levels of the parent CH121 to borers, rice blast, and brown planthopper were S, S, and HS, whereas those of CH121TJH were R, R, and HR. Introduction of *CRY2A*, *Pib*, *Pikm*, and *Bph29* into CH891TJH and R205XTJH had a similar effect on resistance. The resistance levels to borers, rice blast, and brown planthopper were S, S, and S in CH891 and R205X. By contrast, these levels were HR, HR, and R in CH891TJH and R, R, and R in R205XTJH. Differences in resistance to the same pest or pathogen in plants with different genetic backgrounds may be related to the different expression levels of resistance genes or to the different recovery rates of their genetic backgrounds.

Agronomic traits of the resistant strains were similar to those of their recurrent parents, although there were some differences. Thousand-grain weight was slightly lower in the resistant strains, although this difference was only significant for R205XTJH. Resistant strains also tended to be taller, although again this difference was not significant. In general, introduction of resistance genes had relatively minor effects on the agronomic characters of the improved strains; only R205XTJH showed significant reductions in more than one trait, and CH891TJH showed no significant reductions in any trait.

The two varieties CH121TJH and CH891TJH with actual recovery rates higher than the theoretical recovery rate showed good performance in inheriting the resistance of the donor parents, and in only one case did an agronomic trait differ from that of the original parents (total grain number in CH121TJH). R205XTJH, whose actual recovery rate was slightly lower than the theoretical rate, also inherited the resistance of the donors parent but differed from the recurrent parent in several agronomic traits.

## 5. Conclusions

In this study, stem borer resistance gene *CRY 1C/CRY 2A* is transgene, rice blast resistance genes *Pikm* and *Pib*, and the brown planthopper resistance gene *Bph 29* are cis-genes. After testing the resistance of new lines CH121TJH, CH891TJH, and R205 XTJH, we conducted a series of experiments to determine whether the introduction of transgenic rice will cause genetic variation. The results showed that the introduction of transgenic *CRY 1C/CRY 2A* may lead to some differences in agronomic characters. How to reduce the occurrence of genetic variation and improve crop resistance is our future goal. Perhaps it would be a good choice to develop more cis-genes. Molecular marker-assisted selection and backcross techniques were used to simultaneously improve resistance to borers, rice blast, and brown planthopper in three restorer lines. The genetic background and genotype of the improved lines were very similar to those of the parents because of the use of molecular marker-assisted selection, which enabled backcross breeding and reduced the demands for labor and time. The resulting genetic materials can serve as a foundation for future breeding efforts.

## Figures and Tables

**Figure 1 genes-14-01070-f001:**
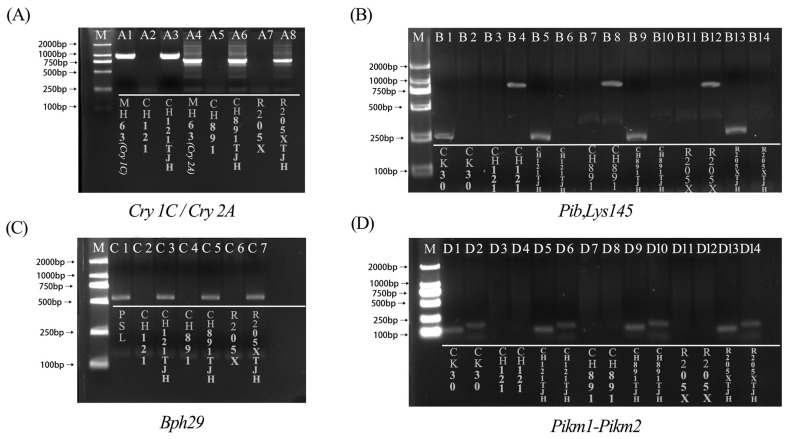
PCR amplification products from donor parents, recurrent parents, and transgenic rice with polymerized resistance genes. The names of the plants in each tunnel are marked on the diagram. M is DL2000 Maker. (**A**). *Cry1C* and *Cry2A*. (**B**). *Pib* and *Lys145*. (**C**). *Bph29*. (**D**). *Pikm1 and Pikm2*. The primer of A1–A3 is *CRY 1C*. The primer of A4–A8 is *CRY 2A.* The primer of B1, B3, B5, B7, B9, B11, and B13 is *Pib.* The primer of B2, B4, B6, B8, B10, B12, and B14 is *Lys145.* The primer of C1–C7 is *Bph29*. The primer of D1, D3, D5, D7, D9, D11, and D13 is *Dkm1*. The primer of D2, D4, D6, D8, D10, D12, and D14 is *Dkm2*.

**Figure 2 genes-14-01070-f002:**
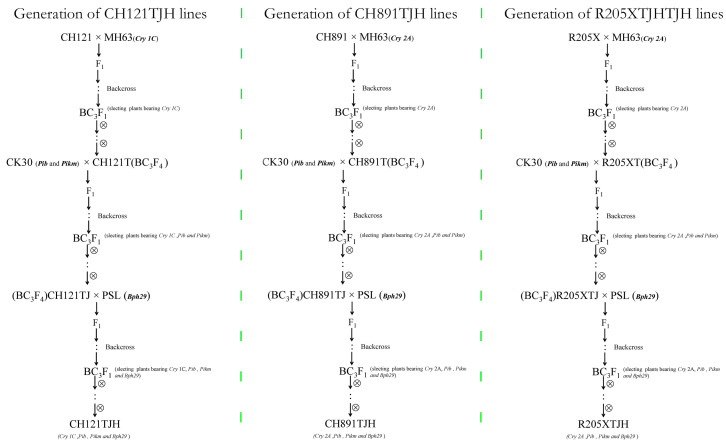
Flow chart depicting the breeding process of polymeric resistance gene lines. CH121TJH was produced by backcrossing MH63 (*Cry1C*) as a donor parent with CH121 as a recurrent parent. CH891TJH was produced by backcrossing MH63 (*Cry2A*) with CH891, and R205XTJH was produced by backcrossing MH63 (*Cry2A*) with R205X. ‘×’ means crossbreeding, ‘⊗’ means self-cross breeding.

**Figure 3 genes-14-01070-f003:**
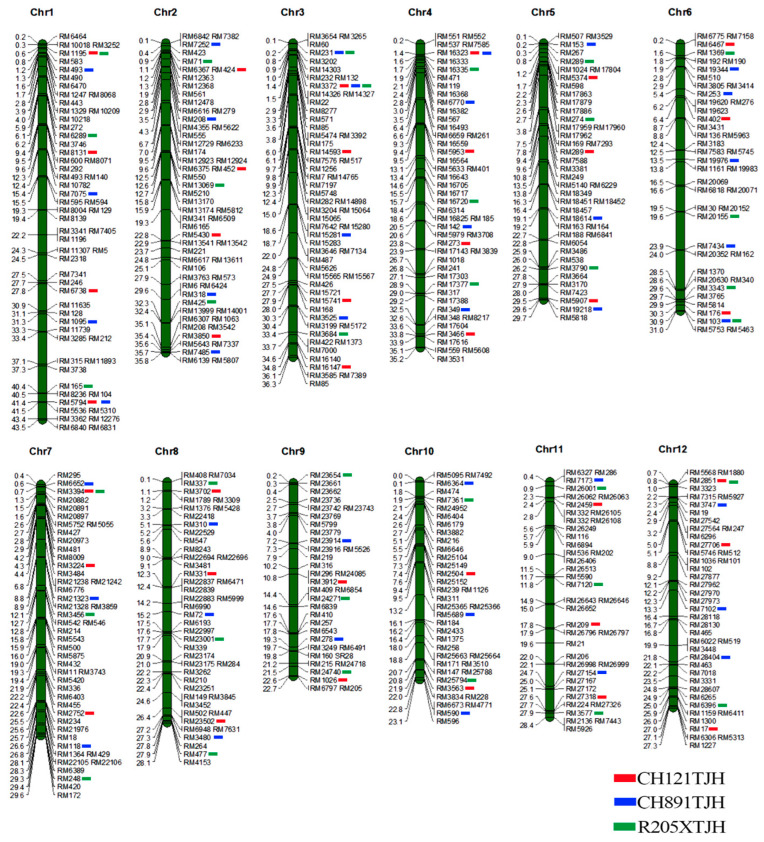
The chromosomal distributions of the 560 microsatellite markers used for genotyping CH121TJH, CH891TJH, R205XTJH, and their recurrent parent. Blue, green, and red bars represent the homozygous markers for the PSL genome in the three restorer lines CH121TJH, CH891TJH and R205XTJH, respectively. The remaining markers were homozygous for the genome of the recurrent parent. The scale on the left indicates the physical position (Mb) of each marker, and the names of the microsatellite markers are on the right.

**Figure 4 genes-14-01070-f004:**
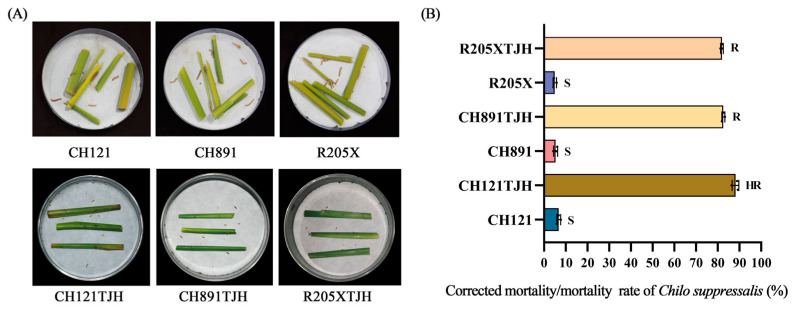
(**A**) Laboratory insect feeding tests using stems of CH121, CH891, R205X, CH121TJH, CH89TJH, and R205XTJH collected at the heading stage. (**B**) Larval mortality of *Chilo suppressalis* in laboratory bioassays. All tests were performed with ten replicates, and one replicate comprised 20 s instar larvae. Values are mean ± standard error.

**Figure 5 genes-14-01070-f005:**
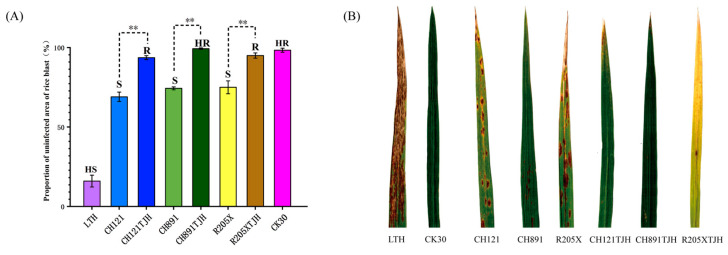
(**A**) Proportion of uninfected leaf area in multiple rice lines after spraying with an *M. oryzae* spore suspension (** *p* < 0.01, LSD test). (**B**) Leaf phenotypes after *Magnaporthe oryzae* inoculation.

**Figure 6 genes-14-01070-f006:**
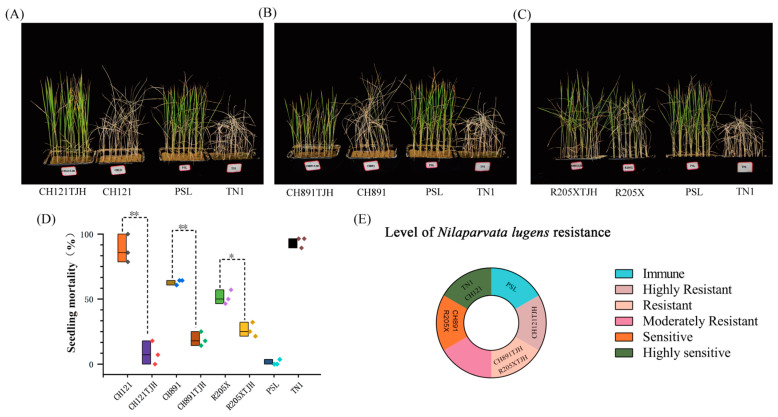
*Nilaparvata lugens* resistance in various parental and transgenic rice lines at the seedling stage. (**A**) CH121TJH. (**B**) CH891TJH. (**C**) R205X. (**D**) Seedling mortality. Medians are indicated by solid bold lines. * 0.01 < *p* < 0.05 (LSD), ** *p* < 0.01 (LSD). (**E**) Pie chart showing the resistance levels of various lines to *N. lugens*.

**Figure 7 genes-14-01070-f007:**
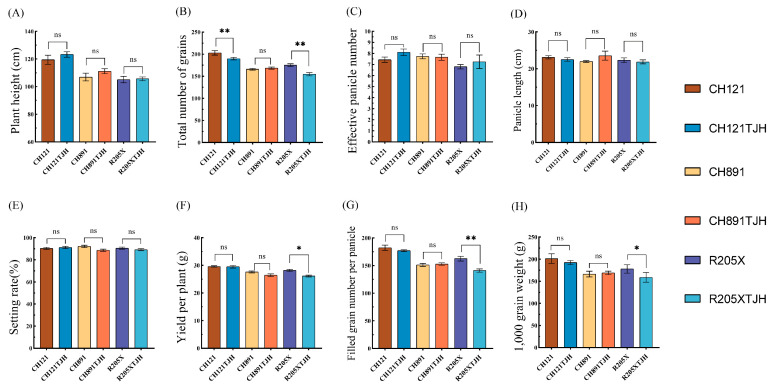
Comparison of eight agronomic traits in CH121TJH, CH891TJH, and R205XTJH with those of their original parents. (**A**) Plant height (cm). (**B**) Total grain number. (**C**) Effective panicle number. (**D**) Panicle length (cm). (**E**) Seed setting rate. (**F**) Yield per plant (g). (**G**) Number of solid grains. (**H**) 1000-grain weight (g). ** *p* < 0.01 (LSD),* 0.01 < *p* < 0.05 (LSD), ns means none otherness.

**Table 1 genes-14-01070-t001:** Plant materials description.

Varity	Type	Provided by	R.Gene	Resistance
MH63(*CRY1C*)	Donor parents	Huazhong Agricultural University, China	*CRY1C*	*Chilo suppressalis*
MH63(*CRY2A*)	Donor parents	Huazhong Agricultural University, China	*CRY2A*	*C. suppressalis*
CK30	Donor parents	Sichuan Agricultural University, China	*Pib* *Pikm*	*Magnaporthe oryzae*
PSL	Donor parents	Chengdu Institute of Biology, China	*Bph29*	*Nilaparvata lugens*
CH121	Acceptor parents	Jiangxi Agricultural University, China	/	/
CH891	Acceptor parents	Jiangxi Agricultural University, China	/	/
R205X	Acceptor parents	Jiangxi Agricultural University, China	/	/

## Data Availability

The original contributions presented in the study are publicly available. The data sets supporting the results of this article are included within the article and its Appendix A. All plant experiments and all field experiments are performed in Jiangxi Agriculture University.

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
