# Peer review of "Stacking Multiple Genes Improves Resistance to Chilo suppressalis, Magnaporthe oryzae, and Nilaparvata lugens in Transgenic Rice"

_genes, 2023, doi:10.3390/genes14051070_

Round 1

Reviewer 1 Report

In the manuscript, Li et al stacked multiple genes into the rice genome for tolerance to several pests and diseases. The authors reported an interesting combination of resistance genes that are certainly of use. However, the manuscript is not easy to follow. Therefore I recommend a thorough revision.

In the title, I think the term polymerization is not appropriate term, I would suggest the title: “Stacking multiple genes improves Resistance to Chilo suppressalis, Magnaporthe oryzae, and Nilaparvata lugens in Transgenic Rice”.

Please also replace the term “gene polymerization” with “gene stacking” throughout the manuscript.

In the abstract, it is unclear what the final product i.e what gene combinations are made and tested, and better performed? Please make it clear which are the transgene and which are natural ones. In line 17-18, did you introduce Cry genes into another transgenic parent? Please make it clear to readers. I am sure that the authors are experts in breeding works. However, making their work easily understandable to readers is equally important.

Ln 39, “Most cultivated rice varieties are single-gene resistant varieties with strong specific resistance.” I think this is not so correct. Write “Most cultivated rice varieties were bred for strong single resistance traits” instead.

Ln 45, please use ‘express’ instead of ‘clone’.

Ln 74-87, not linked to transgenic breeding and could be removed.

Ln 100, please provide more information on the transgenes- CRY1C, CRY2A (promoter coding sequence, copy number, generation etc).

Graphics and figures of high quality must be provided

2.1 Plant materials: The description of genes, sources, background etc seems not easy to follow. I suggest providing them with a table.

4. The first paragraph in the discussion section was already explained in the introduction section. Please shorten or eliminate.

Please discuss your result in light of similar published literature that described stacking multiple (trans)genes in the discussion section.

Please rewrite the conclusion clarifying transgene and cis-gene.

Author Response

Dear reviewer:

Thank you so much for your advice. I will answer your questions and suggestions here.

  1. We revised the title “Stacking multiple genes improves Resistance to Chilo suppressalis, Magnaporthe oryzae, and Nilaparvata lugens in Transgenic Rice”. Thank you very much for your advice.
  2. We have replaced the term “gene polymerization” with“ gene stacking” throughout the manuscript.
  3. We have revised the abstract and explained the genes introduced by each  lines and the properties of the genes. Thank you very much for your suggestion. Our previous statement was indeed unreasonable.
  4. We have revised the contents of line 39, thank you for your suggestion.
  5. We have replaced ‘express’ in 45 lines.
  6. We have removed Ln 74-87.
  7. We added information about CRY1Cand CRY2A in the supplementary materials. We provide a schematic diagram of the vector construction of Cry1C and Cry2A and a schematic diagram of the flanking sequences of Cry1C and Cry2A. Cry1C and Cry2A are provided by Huazhong Agricultural University. They takes wild BT gene as the blueprint, and optimizes the nucleotide sequence based on the codon preference of rice without changing the amino acid sequence. Due to patent reasons, we have not been provided with the corresponding Ubiquitin promoter sequence. We are only allowed to use MH63(CRY1C) and MH63(CRY2A) carrying Cry1C and Cry2A as donor parents, and introduce Cry1C and Cry2A genes into our rice through cross breeding. Therefore, we can't provide you with the exact sequence of Ubiquitin promoter.
  8. We replaced the pictures in the manuscript with high-quality pictures.
  9. Insection1, we added a new table to explain the source, genes and background of plant materials. Thank you for your suggestion. Now the data content will be more intuitive and easy to understand.
  10. We have cut out some redundant and repetitive contents in the discussion. Thank you for your suggestion.
  11. We added new discussion content to the manuscript according to tacking multiple (trans)genes. Thank you for your suggestion.
  12. We rewrite the conclusion clarifying transgene and cis-gene.

Thank you again for your suggestions and questions.

Yours sincerely,

Authors

Reviewer 2 Report

The use of molecular breeding for resistance to insect and fungus pathogens in rice is an important work to develop the resistance variety of the rice which will reduce the use of insecticides and pesticides. The following issues need to be addressed before considering for publication.

Check with the instruction of the authors for the format of the journal.

The data presentation needs to be improved extensively. There is a lot of confusion for the readers in the present form. 

-        Italics pathogen name in abstract and all over manuscript.

-        The significance of the mentioned insects and fungus is missing in the introduction, and the data on what percentage of qualitative and quantitative loss occur due to these pathogens.

-        Line#55 what is meant by Plant DNA virus-mediated transformation?

-        Line#65 what is meant by good strain? The use of scientific language will be more appropriate

-        There is no explanation of why the  CH121; CH891 and R205X were used as recipient plants?

-        Reference for CTAB method?

-        Line 143 Previous work…. Would be a better fit for the result section

-        A table either in supplementary mentioning the name of the gene/protein with amplicon size would be better. As Line 140-146 is confusing.

-        Reference for section 2.3 is missing

-        How age synchronization of the larvae was managed?

-        What was the initial number of larvae per petri plate in section 2.4?

-        Use the complete form of no abbreviation if used for the first time in the manuscript.

-        What was the concentration of spores that was used in section 2.5?

-        CH121 and CH891 are sensitive to their respective pathogen but the yield data shows that there is no significant difference between the parent line and the CH121TJH and CH891TJH. What would be the better explanation for this? Because if there is no effect on the crop yield then what was the purpose of all the hard work?

Author Response

Dear reviewer:

Thank you so much for your advice. I will answer your questions and suggestions here.

  1. We have Checked with the instruction of the authors for the format of the journal and the format has been modified. Thank you for reminding us.
  1. We improved the data presentation, deleted some unnecessary data and supplemented some data.
  2. The italics of pathogen names in abstracts and all manuscripts have been revised.
  3. We add the significance of the mentioned insects and fungus in the introduction.
  4. In line#55 we mentioned Plant DNA virus-mediated transformation. The propensity of viruses to acquire genetic material from relatives and possibly from infected hosts makes them excellent candidates as vectors for horizontal gene transfer. The transformation method mediated by plant DNA virus is a means of transferring foreign genes. We give an example here to illustrate the commonly used transformation methods, although this method is not used in this experiment.
  5. We revised the contents of line#65.Thank you very much for your advice.
  6. CH121, CH891 and R205X were used as recipient plants because they have good adaptability in the middle and lower reaches of the Yangtze River, the main rice producing area in China. We supplemented this content in the manuscript and used a table to describe the source and characteristics of each parents.
  7. We really didn't reference CTAB method in the previous manuscript. Thank you very much for reminding us. Now it has been added.
  8. Line#143 Previous work...wanted to explain detection of PCR products, Pikm gene can be successfully introduced only when two bands appear, so we think it is more appropriate to explain this part in section 2.2.
  9. Thank you very much for your proposal. We have made a new table (Table S2) to explain the information of amplified genes in the supporting Information.
  10. In section 2.3, we did not refer to the source of the calculation formula. Thank you very much for reminding us. Now we have added a reference.
  11. We let the female adults of Chilo suppressalis lay eggs after catching them. Each female adult can provide about 300 eggs, and at least 100 can be caught at a time. The hatching rate of eggs is about 70%. We hatch the eggs laid at the same time, so that we can ensure that the age of larvae is the same.
  12. In section 2.4,initial number of larvae per petri plate is 30.We have now added this explanation to the manuscript. Thank you very much for your patient examination and suggestions.
  13. We have changed all the abbreviated forms to non-abbreviated forms.
  14. In section 2.5,the concentration of spores was about 30x104 conidia/ml.
  15. We used pesticides and fungicides in the cultivation process in order to create an environment to control pests and diseases. The purpose of this is to examine whether there is a huge yield difference between the transgenic rice lines and their corresponding parents under normal conditions without pests and diseases. Because what we want to explain in this study is that the introduction of resistance genes will not affect the yield and other agronomic traits, and the difference between transgenic lines and their parents only exists in the resistance genes rather than the yield. We also added this content in the manuscript. If there are pests or diseases, the yield difference between transgenic lines and their parents is definitely huge, because the transgenic lines are resistant and the parents are not resistant, and the yield of transgenic lines will definitely be higher than that of their parents. If you are interested in the yield problem in the case of insect pests, you can refer to another paper of our team.The title is Comparison of the Phenotypic Performance, Molecular Diversity, and Proteomics in Transgenic Rice. [https://doi.org/10.3390/plants12010156].This paper we let rice exposed naturally to the insect at the tillering stage. And the yield after suffering from insect pests is counted.

Thank you again for your suggestions and questions.

Yours sincerely,

Authors

Reviewer 3 Report

Dear authors,

This is good study to combat agricultural pests. I have no concerns with this study but still would like to know some aspects. 

1. Can a bar graph showing the harvest index of these transgenics be shown? This will give the productivity of these plants.

2. Paper needs formatting. All the headings of the paragraph are not aligned.

3. Apart from biotic stress was any form of abiotic stress treatment given on the plant.

4. Was the functioning of the photosynthetic parameters checked? eg conductance, transpiration rate?

5. What was the transcript level of the genes WT in comparison with the developed lines? How much did it change?

6. So, what might be the causative effect of stacking these two genes that these plants are having and providing resistance? Please discuss in the discussion section.

7. Was any other pest or insects tested on these developed transgenic lines out of curiosity?

Author Response

Dear reviewer:

   Thank you so much for your advice. I will answer your questions and suggestions here.

  1. Figure 7. showing the harvest index of three transgenic lines. We shown total grain number, effective panicle number, seed setting rate, yield per plant, number of solid grains and 1000-grain weight in this figure. These materials can describe the harvest index of three transgenic lines. In this study, we are mainly committed to explaining that there is little or no difference in yield between the improved transgenic lines and the original parents, thus proving that the introduction of resistance genes has little effect on yield after improvement rather than emphasizing the improvement of yield. Because we control the influence of pests and germs on the cultivation conditions. In the environment without pests and diseases, it is normal that the yield of improved lines is similar to that of their parents.
  2. We have formatted the paper, thank you for reminding us.
  3. We didn' t design the experiment of abiotic stress treatment.Your suggestion gives us a new idea. In recent years, high temperature and dry weather occurred frequently in the experimental area. We have plans to carry out experimental treatment under dry conditions next. Thank you very much for your opinion.
  4. We didn't checked the function of the photosynthetic parameters. In this study, we mainly conducted resistance identification and yield analysis on the improved lines. Your suggestion is very meaningful. We will test the data you mentioned in the next study on drought management.
  5. In WT lines, we analyzed PCR amplification products. As you can see from the results in Figure 1, there are no bands in the WT strain because there is no R gene or the expression of R gene is not high. Therefore, we decided not to study the transcriptlevel of the WT strain.If you are interested in the study of transcript level, another paper of our team is about this aspect. The title is Analysis of the Genetic Stability of Insect and Herbicide Resistance Genes in Transgenic Rice Lines: A Laboratory and Field Experiment.      DOI: 10.1186/s12284-023-00624-5      [https://pubmed.ncbi.nlm.nih.gov/36781713/]
  6. We have discussed in the discussion section about resistance of three improved These resistances are inherited by improved lines from their donor parents.
  7. We haven’t use other pest or insects tested on these developed transgenic lines. But we have this plan next. Thank you very much for your advice.

   Thank you again for your suggestions and questions.

Yours sincerely,

Authors

Round 2

Reviewer 2 Report

The authors have improved the manuscript as suggested by the reviewer. It can be accepted for publication in the GENES.